# β-Sheet to Random Coil Transition in Self-Assembling Peptide Scaffolds Promotes Proteolytic Degradation

**DOI:** 10.3390/biom12030411

**Published:** 2022-03-07

**Authors:** Elsa Genové, Nausika Betriu, Carlos E. Semino

**Affiliations:** Tissue Engineering Research Laboratory, Department of Bioengineering, IQS-School of Engineering, Ramon Llull University, 08017 Barcelona, Spain; egenove@vetilea.com (E.G.); nausikabetriur@iqs.url.edu (N.B.)

**Keywords:** self-assembling peptides, RAD16-I, scaffold, degradation, proteolysis, β-sheet, random coil

## Abstract

One of the most desirable properties that biomaterials designed for tissue engineering or drug delivery applications should fulfill is biodegradation and resorption without toxicity. Therefore, there is an increasing interest in the development of biomaterials able to be enzymatically degraded once implanted at the injury site or once delivered to the target organ. In this paper, we demonstrate the protease sensitivity of self-assembling amphiphilic peptides, in particular, RAD16-I (AcN-RADARADARADARADA-CONH_2_), which contains four potential cleavage sites for trypsin. We detected that when subjected to thermal denaturation, the peptide secondary structure suffers a transition from β-sheet to random coil. We also used Matrix-Assisted Laser Desorption/Ionization-Time-Of-Flight (MALDI-TOF) to detect the proteolytic breakdown products of samples subjected to incubation with trypsin as well as atomic force microscopy (AFM) to visualize the effect of the degradation on the nanofiber scaffold. Interestingly, thermally treated samples had a higher extent of degradation than non-denatured samples, suggesting that the transition from β-sheet to random coil leaves the cleavage sites accessible and susceptible to protease degradation. These results indicate that the self-assembling peptide can be reduced to short peptide sequences and, subsequently, degraded to single amino acids, constituting a group of naturally biodegradable materials optimal for their application in tissue engineering and regenerative medicine.

## 1. Introduction

Tissue engineering and regenerative medicine require biomaterials capable of maintaining and guiding cell growth, migration, differentiation and function. The ideal biomaterial for cell scaffolding should mimic the native extracellular matrix (ECM) and should also allow the diffusion of nutrients, oxygen and metabolic waste products. In addition, the scaffold should only serve as a temporary support, and ideally it should be gradually degraded while cells build their own ECM [1]. Scaffolds from natural origins are mainly hydrogels that have the advantage of resembling the in vivo ECM since they present with biological binding sites and are also susceptible to degradation by the cells [2], which allows both ECM turnover and cell migration within the scaffold. However, they show some limitations, such as batch-to-batch variability, unquantified constituents and impurities; a fact that may compromise assay reproducibility [3]. They can also be extremely instructive and provide too much signaling to the cells, which in some applications might be a drawback as this could induce an undesired phenotype. Thus, currently, there is significant interest in and efforts focused on designing and synthesizing completely defined materials that increasingly mimic the complexity of the extracellular matrix in terms of structure, cellular biorecognition and function, and degradability specific for each tissue of interest.

Regarding the biodegradation of the scaffold, many synthetic biomaterials have been designed in a way to suffer degradation by ester hydrolysis [4], which is in contrast with the in vivo ECM turnover that is degraded and remodeled mainly by matrix metalloproteinases (MMP) [5,6]. Synthetic polymers such as poly-lactic acid (PLA) [7], poly-glycolic acid (PGA) [8] and their copolymers, poly-caprolactone (PCL) [9] and poly(propylene fumarate) (PPF) [10] are typical examples of synthetic scaffolds that suffer hydrolytic degradation. These polymers can be synthesized and tuned varying the molecular weight and glycolic acid:lactic acid ratio to assess the desired degradation rate. However, their degradation products may lead to the formation of an acidic local microenvironment around the area of degradation which may cause damage to the surrounding cells [11]. Currently, special efforts and progress is being made towards the synthesis of enzyme-sensitive biomaterials, containing recognition sites for proteases such as matrix metalloproteinases (MMPs) [12,13,14] and serine proteases such as urokinase [15] plasmin [16], elastase [17] or trypsin [18], which could be used for tissue engineering as well as for drug delivery purposes. For example, trypsin-recognized hydrogels could be useful for encapsulating drugs for delivery into the small intestine [18] while the design of scaffolds with MMP-cleavable sequences could be relevant for the development of anti-cancer drug delivery platforms, as matrix metalloproteinases are highly overexpressed in tumor tissues [19,20]. Moreover, the importance of MMP-mediated degradation during capillary morphogenesis justifies the presence of MMP-degradable peptide sequences in synthetic hydrogels used to promote and study vasculogenesis [16]. Additionally, the presence of protease-sensitive domains in hydrogels used for tissue engineering applications may confer scaffold biodegradability, thus, avoiding the need for scaffold removal after tissue regeneration [17].

Our group has vast experience working with amphiphilic self-assembling peptides. In particular, we have investigated RAD16-I (AcN-RADARADARADARADA-CONH_2_, R arginine, A alanine, D aspartic acid) as a candidate biomaterial for tissue engineering and regenerative medicine [21]. This synthetic scaffold presents with a β-sheet structure and self-assembles into a network of interweaving nanofibers of ~10 nm in diameter, forming hydrogel scaffolds with pore sizes ranging from 50 to 200 nm and more than 99% water content [22]. This peptide scaffold has been demonstrated to promote cell adhesion [23], phenotype maintenance [24], differentiation [24,25] and proliferation [26,27] of a variety of mammalian cells. For example, RAD16-I has been proven to support chondrogenic [28], osteogenic [29], adipogenic [29] and cardiac [30] differentiation of adult stem cells. Moreover, the culturing of rat liver-derived progenitor cells in RAD16-I promoted their differentiation into functional hepatocytes [31]. Furthermore, this peptide scaffold has been modified by adding biologically active sequences from proteins of the extracellular matrix such as laminin and fibronectin to modulate cell response [24,32,33]. RAD16-I is commercially available for research use as PuraMatrix^TM^ (3D-Matrix, Tokyo, Japan) (1% in water). Other related products are PuraStat^®^ (3D-Matrix, Tokyo, Japan) and PuraBond^®^ (3D-Matrix, Tokyo, Japan) (both at 2.5% aqueous RAD16-I formulation), which are CE-marked as Class III medical devices for hemostatic use in humans, and PuraSinus^®^ (3D-Matrix, Tokyo, Japan) (also at 2.5% in water), which was cleared by the FDA in 2019 as an intraoperatively applied wound dressing [34]. It has been reported that over time, applied PuraStat^®^ and PuraSinus^®^ are degraded to their constituent natural l-amino acids by endogenous proteolytic/hydrolytic mechanisms, which are then metabolized or recycled, removing all residual RAD16-I from the surgical site. The proteolytic degradation of RAD16-I has been the subject of many questions by researchers working in the field, but to date, digestion susceptibility of this peptide in vitro has not been assessed.

In this paper, we propose enzymatic digestion assays to demonstrate the protease sensitivity of the self-assembling peptide RAD16-I. We focused the research on the tryptic degradation of RAD16-I because its peptide sequence (AcN-RADARADARADARADA-CONH_2_) has four potential cleavage sites for trypsin and this widely studied serine-protease is known to cleave at the C-terminal of Arginine (R) and Lysine (K) [35]. In the first approach, we studied the thermal stability of this self-assembling peptide scaffold. We used circular dichroism to detect secondary structure changes in the peptide scaffold when subjected to thermal denaturation, and we observed a transition from β-sheet to random coil structures of diluted samples of the peptide. We also used MALDI-TOF to detect the proteolytic-breakdown products of samples subjected to trypsin incubation and atomic force microscopy (AFM) to visualize the effect of the degradation on the scaffold nanofibers. Results showed that samples that underwent thermal denaturation had a higher extent of degradation than non-denatured samples, suggesting that the transition from β-sheet to random coil leaves the cleavage sites accessible and susceptible for protease degradation.

## 2. Materials and Methods

### 2.1. Circular Dichroism (CD)

Circular dichroism studies were performed on a JASCO-810 spectropolarimeter equipped with a Peltier system. RAD16-I samples (commercially available as PuraMatrix^TM^, 354250, Corning, Corning, NY, USA) were diluted from a peptide stock solution (1% in water, 5.38 mM) in deionized water to a final concentration of 25, 50 and 75 µM and allowed to equilibrate at 20 or 90 °C for 10 min prior to analysis. CD data were acquired in a range from 195–240 nm at a band width of 1 nm and scan speed of 20 nm/s.

### 2.2. Atomic Force Microscopy (AFM)

To image the thermal denaturation of the self-assembling peptide, RAD16-I samples were diluted in water from a stock solution (5.38 mM) to a final concentration of 100 µM. The thermal treatment consisted of subjecting the sample to a 90 °C incubation for 10 min. Then, 1 µL of sample was deposited onto freshly cleaved mica (Grade V5 mica, Structure Probe Inc., 01804-CA, West Chester, PA, USA), rinsed twice with Milli-Q water and observed under the AFM. For RAD16-I degradation experiments, samples were prepared as follows: RAD16-I was diluted in water from a stock solution at a final concentration of 100 µM. For thermal treated samples, a 10 min denaturation step (incubation at 90 °C) was performed prior to the enzymatic degradation. Samples were cooled at room temperature and trypsin (Merck, T1426) was added at a 1:1000 molar ratio (enzyme:peptide). Samples (1 µL) were taken at 10 and 60 min and were deposited onto freshly cleaved mica, rinsed twice with Milli-Q water, air dried and scanned under the AFM.

AFM images were obtained with a silicon-scanning probe (AC240-TS, Asylum Research) with a resonance frequency of 70 KHz, spring constant 2 N/m, tip curvature radius <10 nm and 240 μm length. Images were obtained with a Multimode AFM microscope (Nanoscope IIIa, Digital Instruments, Santa Barbara, CA, USA) operating in TappingMode. AFM scans were taken at a resolution of 512 × 512 pixels and produced topographic images of samples, whose brightness of features increased as a function of height. Typical scanning parameters were as follows: tapping frequency 75 kHz, RMS amplitude before engagement 1–1.5 V, setpoint 0.7–1 V, integral and proportional gains from 0.3–0.6 and 0.4–0.6, respectively, and scan rate 1.51 Hz.

### 2.3. MALDI-TOF

Matrix-assisted laser desorption/ionization time of flight (MALDI-TOF) of trypsin-digested peptide samples was performed on a Bruker Ultraflex mass spectrometer, equipped with a pulsed nitrogen laser (337 nm). The peptide RAD16-I was diluted from a stock solution to a final concentration of 100 µM in water. Trypsin was added in a molar ratio 1000:1 (peptide:enzyme). Peptide samples were incubated with trypsin at 37 °C for 0, 1, 5, 10 and 30 min (performed in duplicate), and boiled in order to inactivate trypsin. Samples were prepared by mixing 0.5 µL of the peptide solution with 1 µL of the matrix solution (0.3 mg/mL, α-cyano-4-hydroxy-cinnamic acid in ethanol/acetone 2:1). The mixture was then spotted on an anchor-chip target (Bruker, Billerica, MA, USA) and allowed to evaporate to dryness at room temperature. Spectra were acquired in positive reflectron mode, using an acceleration voltage of 25 kV. Acquisition was performed in automated mode, using the Autoexecute software (Bruker, Billerica, MA, USA). A total of 1000 spectra, collected in 50-shot series at different positions of the sample spot were averaged for each sample. A mixture of standard peptides of known molecular mass in the range from 750–3500 Da (Bruker) was used for external calibration of the mass spectra.

## 3. Results

### 3.1. Structural Characterization of the Self-Assembling Peptide RAD16-I

As a self-assembling peptide, RAD16-I is composed of repeating units of hydrophilic and hydrophobic amino acids, in which the charged residues alternate positive and negative charges, forming a β-sheet structure in aqueous solutions (Figure 1). This characteristic sequence is the one responsible for the self-assembly of the peptide molecules into a hydrogel, formed by a network of interweaving nanofibers from 10–20 nm in diameter and 50–200 nm in pore size and, thus, the applicability of the material as a scaffold in biomedical applications [36]. Circular dichroism (CD) studies were performed to characterize the RAD16-I peptide structure [37]. As a peptide with a β-sheet secondary structure, RAD16-I circular dichroism spectrum at room temperature (20 °C) presented a minimum molar ellipticity at 218 nm, which represents the β-sheet content, and a maximum of approximately 195 nm, which corresponds to the backbone twist of the peptide in the β-sheet configuration [38]. This characteristic β-sheet spectrum was obtained when the peptide was diluted in water at different concentrations of 25, 50 and 75 µM (Figure 2a–c). This spectrum was also maintained when the peptide was incubated at 40 °C. However, when heated at 90 °C, the peptide suffered a structural transition from β-sheet to random coil, demonstrated by a shift in the CD spectrum (Figure 2a–c) from a maximum of 195 nm for a β-sheet structure to a minimum in this same wavelength for the random coil configuration. Moreover, this shift was shown to be irreversible at room temperature for at least a week. Surprisingly, the thermal denaturalization was not observed when concentrated solutions of the peptide (5 mg/mL in water, 2.9 mM) were incubated at different temperatures, indicated by the maintenance of the standard β-sheet spectrum in the samples incubated both at 20 °C as well as 90 °C (Figure 2d). Therefore, RAD16-I has been shown to exhibit thermal stability at high peptide concentrations (2.9 mM) but not at diluted ones (25–75 µM), suggesting a concentration-dependent resistance to thermal denaturation, which could be explained by a higher self-assembly degree of the peptide molecules.

We next performed AFM imaging to observe the structure of nanofibers in both untreated and thermally treated samples. In untreated samples of a diluted solution, RAD16-I was characterized by a dense network of long nanofibers (Figure 3a,b). On the other hand, the network of nanofibers in thermally treated samples of RAD16-I was significantly less dense, and shorter fibers or fragments could also be detected (Figure 3c,d), which indicates the loss of self-assembling capacity. This loss in the folding of the peptide may be driven by the change in the peptide structure from β-sheet to random coil, as demonstrated with the CD studies (Figure 2).

### 3.2. Enzymatic Degradation of RAD16-I Assessed by MALDI-TOF

We next investigated the protease susceptibility of RAD16-I by MALDI. Trypsin was chosen to perform the experiments as it is a serine protease that cleaves at the C-terminal of arginine (R) and lysine (K) (except when the following amino acid is a proline) [35]. Trypsin is a very common enzyme widely used, especially in molecular biology, and because of its specificity it is used in proteomic studies to perform protein identification. Thus, according to the amino acid sequence of the peptide (Figure 4a) we expected to find the peptide fragments described in Table 1.

To evaluate RAD16-I trypsin degradation, we incubated the peptide with trypsin with or without the previous denaturation step (incubation at 90 °C for 10 min). Thus, we incubated the RAD16-I samples with trypsin at 37 °C for different periods of time, and at each timepoint we mixed the reaction product with the α-cyano-4-hydroxy-cinnamic acid matrix and let the sample dry. Finally, samples were analyzed by MALDI. Results show that under control conditions (untreated RAD16-I), there is mainly one peak with *m*/*z* 1712, corresponding to the full RAD16-I peptide (Figure 4b). Furthermore, small peaks with *m*/*z* 1642 and 1558 could be detected, which may correspond to the peptide chain without one amino acid residue, an alanine and an arginine, respectively. These peptides are most likely impurities from the RAD16-I peptide synthesis.

We next aimed to determine if the structural changes observed by CD after denaturing RAD16-I (Figure 2) had any influence on proteolytic degradation. For that, peptide stock solution was diluted to 50 µM and then denatured or not for 10 min at 90 °C. Then, trypsin was added and incubated at 37 °C for different periods of time. The proteolytic reaction was inactivated by boiling the samples, and samples were finally mixed with the matrix and MALDI spectra were acquired. Results show that most of the expected fragments could be found on the trypsin digests (Table 1) in both non-treated (Figure 4c,d) and thermally treated samples (Figure 4e,f), confirming the peptide susceptibility to proteolytic degradation. Moreover, the peak belonging to RAD16-I (*m*/*z* 1712) was significantly reduced when samples were incubated with trypsin, regardless of the incubation time (1 min or 30 min), which had no effect on the relative intensity peak profile (Figure 4c,d). Similarly, the incubation time of the peptide with trypsin did not affect the intensity peak profile obtained for thermally treated samples (Figure 4e,f). Moreover, no differences between untreated and thermally treated samples were detected. Finally, to confirm that incubation time with trypsin did not affect the peak profile, we plotted the evolution of the peaks as a function of incubation time with trypsin (Figure 4g). As expected, no differences were detected in any of the peaks studied, which is a semi-quantitative manner to confirm what was observed with the spectra.

We hypothesized that these unexpected results were due to an inefficient activation of trypsin when MALDI samples were prepared, in which trypsin was inactivated by boiling the samples. In order to prove this hypothesis, the digestion was performed again in thermally treated and untreated RAD16-I samples. In this case, the samples were taken and immediately mixed with the MALDI-TOF matrix and evaporated on the sample holder. Results show that the degradation of the peptide is very fast in both cases, but differences in the *m*/*z* 1713 peak at the initial minutes can be detected (Figure 4h), suggesting that the peptide secondary structure influences proteolytic degradation.

### 3.3. Enzymatic Degradation of RAD16-I Assessed by AFM

Because MALDI spectra results indicated peptide proteolysis upon trypsin treatment (Figure 4), we expected to detect structural changes in the peptide using AFM analysis. RAD16-I samples incubated with trypsin were taken at 10 and 60 min and directly spotted into freshly cleaved mica. Then the mica was rinsed with deionized water, dried and observed under the AFM. In this case, strong differences were found between untreated (Figure 5a,b) and thermally treated samples (Figure 5c,d). In particular, in control samples of RAD16-I, a slight change in nanofiber density was observed from 10 min (Figure 5a) to 60 min (Figure 5b) trypsin incubation times. When samples were subjected to thermal treatment and then incubated with trypsin for 10 min (Figure 5c), a dramatic change in the nanofibers was observed, as only the presence of short fibers was detected. We also noticed the presence of cross-ramification corners, which may be less accessible to the enzyme. Interestingly, at longer incubation times (60 min), most of the fibers had disappeared (Figure 5d). This discrepancy between the results obtained with both techniques (MALDI and AFM), reinforces the hypothesis of an inefficient inactivation of trypsin when MALDI samples were prepared. On the contrary, when samples are spotted on freshly cleaved mica, the proteins are immobilized on the surface of the mica and trypsin can no longer perform its function. The results obtained with AFM analysis (Figure 5), together with the fact that the peak belonging to RAD16-I (*m*/*z* 1713) disappears from the MALDI spectra upon trypsin treatment much faster in denatured samples (Figure 4h), demonstrate that tryptic degradation is influenced by the secondary structure of RAD16-I (Figure 6).

## 4. Discussion

In this paper, some physicochemical features of the synthetic self-assembling peptide RAD16-I have been studied. This peptide is currently commercialized under the name of PuraMatrix^TM^ and is being used as a scaffold in many biomedical applications and studies such as in the development of in vitro cancer models [27,39,40], in different tissue engineering scenarios [29,30,41,42] and it has also been tested as a platform for drug delivery [43,44,45,46]. Thus, in addition to performing all the necessary experiments to characterize the biological systems obtained for each different application, it is also essential to control and study all the properties regarding the biodegradability of the scaffold, which, to date, has not been assessed.

Here, we show that diluted (0.025−0.075 mM) but not concentrated (2.9 mM) solutions of RAD16-I can undergo a thermal denaturation when treated at high temperatures (90 °C), demonstrated by a change of its secondary structure analyzed by CD (Figure 2) and confirmed with AFM (Figure 3). These results suggest that resistance to thermal denaturation strongly depends on peptide concentration, which could be explained by a higher self-assembly degree of the peptide molecules that promotes nanofiber structure stability. Instead, much lower concentrations (between 50- and 100-fold) of peptide solutions enhance the detection of random coil configuration after thermal treatment, which suggests a dramatic change in the relative amount of β-sheet and random coil configurations. Moreover, previous studies have shown that thermal stability of fibrils is related to side chain interactions. For example, in highly hydrophobic peptides, fibrils become more stable with increasing temperature, while for less hydrophobic peptides, fibrils become less stable with temperature increase [47,48]. Moreover, all-atom simulations confirmed that, for amphiphilic sequences where the non-polar residue is alanine (which would be the case of RAD16-I), fibrils become less stable with increasing temperature [49], while the opposite occurs when the non-polar residue is phenylalanine, leucine or valine [49]. Hence, it would be interesting to test this principle in another commonly-used self-assembling peptide, KLD-12 (KLDLKLDLKLDL) [50], in which the non-polar residue is a leucine (L). In this case, one would expect to detect an increase in fibril stability as temperature increases, contrary to RAD16-I.

Moreover, AFM evaluation of trypsin proteolysis at different incubation times (10 and 60 min) in untreated and denatured RAD16-I samples (Figure 5) indicate that the change in secondary structure from β-sheet to random coil facilitates the proteolytic process. The proposed model (Figure 6) suggests that at a lower peptide concentration, the amount of shorter nanofibers increases with the consequential increase of nanofiber ends. It is at these nanofiber ends where β-sheet peptides are in equilibrium with soluble random coil peptides in the solution. In this way, either the reduction in peptide concentration or increase in temperature, or both, increase the transition from β-sheet to random coil peptide configuration. Moreover, results suggest that the loss of β-sheet configuration may leave the cleavage sites accessible for enzymatic attack. Moreover, we hypothesize that the tryptic degradation detected in the control samples (not thermally treated) (Figure 4c,d) may be due to equilibrium in the β-sheet-random coil in the ends of the fibers, which would allow trypsin accessibility to the cleavage sites (Figure 6). This is in agreement with previous experimental models of fibril-monomer equilibrium in β-amyloid proteins and theoretical models of amyloid fibril formation [51,52]. Taken together with these studies, our results demonstrate that (1) for a given amount of β-sheet peptides in the system, the equilibrium concentration (C*) of RAD16-I in random coil configuration in the solution increases with temperature, and (2) if cleavage of RAD16-I by trypsin reduces the concentration of RAD16-I in the random coil below the equilibrium concentration (C*), then, peptides at the end of the nanofiber will detach to reach the equilibrium concentration. This not only explains why there are less nanofibers in the system as the incubation time with trypsin increases, but also predicts that if the incubation time with trypsin increases, even less nanofibers are expected to be in the system.

Trypsin is synthesized by pancreatic acinar cells as an inactive form called trypsinogen, which is cleaved and activated by enteropeptidase when it reaches the small intestine, where it performs its function as a digestive enzyme. In this paper, we have demonstrated for the first time the enzymatic degradability of RAD16-I by trypsin, which expands the possible applications of this peptide in biomedicine. Specifically, given its ability to be degraded by trypsin, RAD16-I would be an excellent delivery platform for drug release into the small intestine. Moreover, RAD16-I has been used and demonstrated to be suitable as a carrier for drug encapsulation and controlled release kinetics [34,43,45,46] from the hydrogel by adjusting the peptide concentration and chemistry [53]. Finally, our results indicate that the self-assembling peptide RAD16-I can be digested into shorter peptide sequences and, subsequently, degraded into amino acids, which are a group of naturally biodegradable molecules. In fact, previous studies suggest that when applied as an intraoperative wound dressing, RAD16-I is naturally removed from the surgical site by endogenous proteolytic mechanisms [34]; however, to our knowledge this is the first time that in vitro proteolysis is demonstrated.

## 5. Conclusions

The results obtained in this study are a clear demonstration that the RAD16-I peptide is susceptible to proteolytic degradation, which is exacerbated after peptide thermal denaturation. This opens the door for future research involving different enzymes, such as elastases or matrix metalloproteinases, to further investigate the proteolytic susceptibility of this self-assembling peptide. Moreover, these results may help in the development of more applications of RAD16-I in biomedicine, as its proteolytic degradation has been the subject of many questions by researchers working in the field.

## Figures and Tables

**Figure 1 biomolecules-12-00411-f001:**
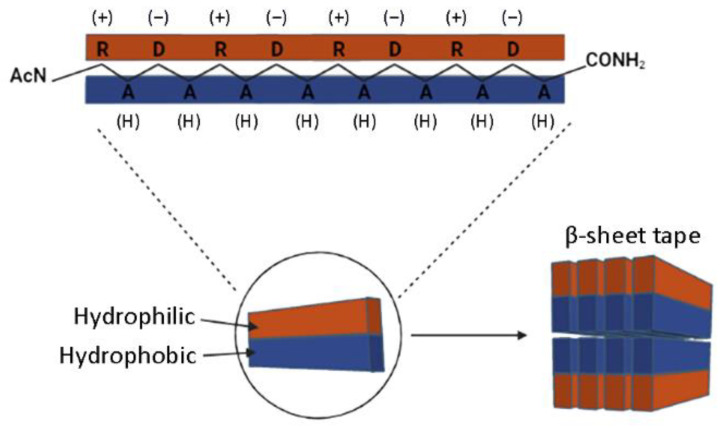
Schematic model of the nanofiber developed by self-assembling RAD16-I molecules. The peptide alternates hydrophilic (R, arginine and D, aspartic acid) and hydrophobic (A, alanine) amino acids that form a β-sheet structure in aqueous solutions. The nanofiber is formed by a double tape of assembled RAD16-I molecules in antiparallel β-sheet configuration.

**Figure 2 biomolecules-12-00411-f002:**
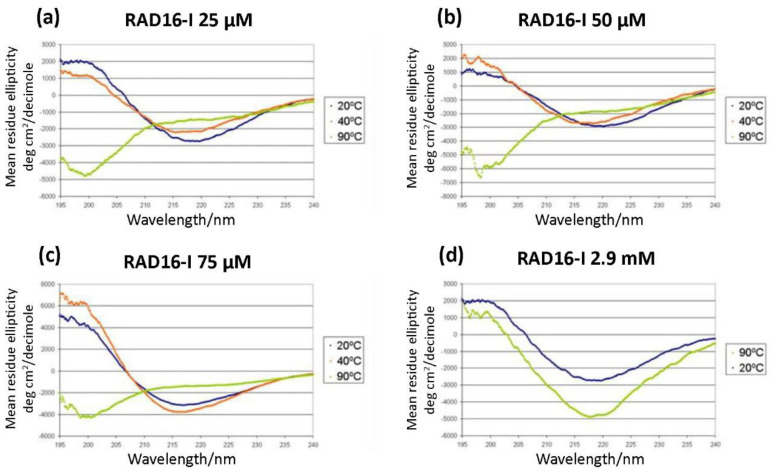
Circular dichroism studies of RAD16-I samples diluted in deionized water at a concentration of (**a**) 25 µM; (**b**) 50 µM; (**c**) 75 µM. Spectra were recorded at different temperatures of 20, 40 and 90 °C; (**d**) Circular dichroism of non-diluted peptide samples (5 mg/mL in water, 2.9 mM). Stock peptide samples (2.9 mM) were incubated at 20 or 90 °C for 10 min before being diluted to 25 µM to record the spectra.

**Figure 3 biomolecules-12-00411-f003:**
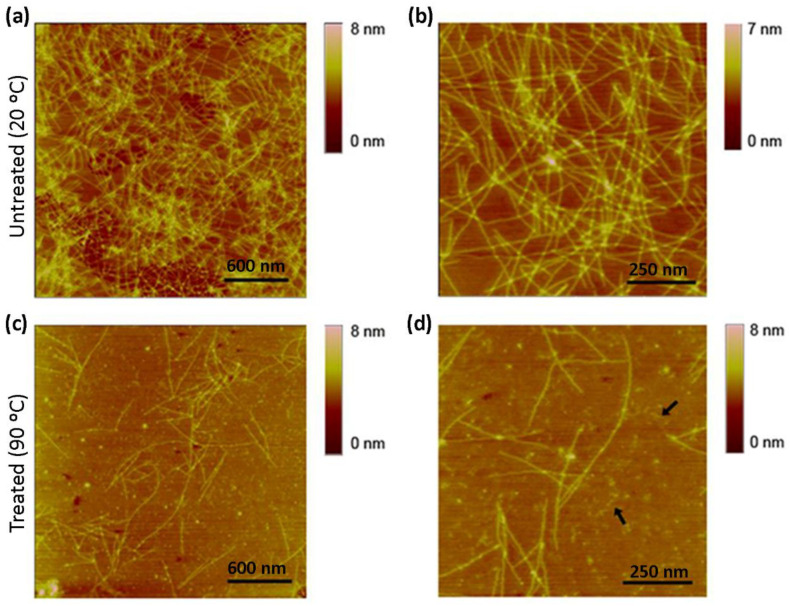
AFM images of the self-assembling peptide RAD16-I. (**a**) RAD16-I nanofibers at room temperature; (**b**) higher magnification images of untreated samples; (**c**) RAD16-I nanofibers after 10 min denaturation at 90 °C; (**d**) higher magnification images of thermally treated samples. Arrows show small nanofiber fragments.

**Figure 4 biomolecules-12-00411-f004:**
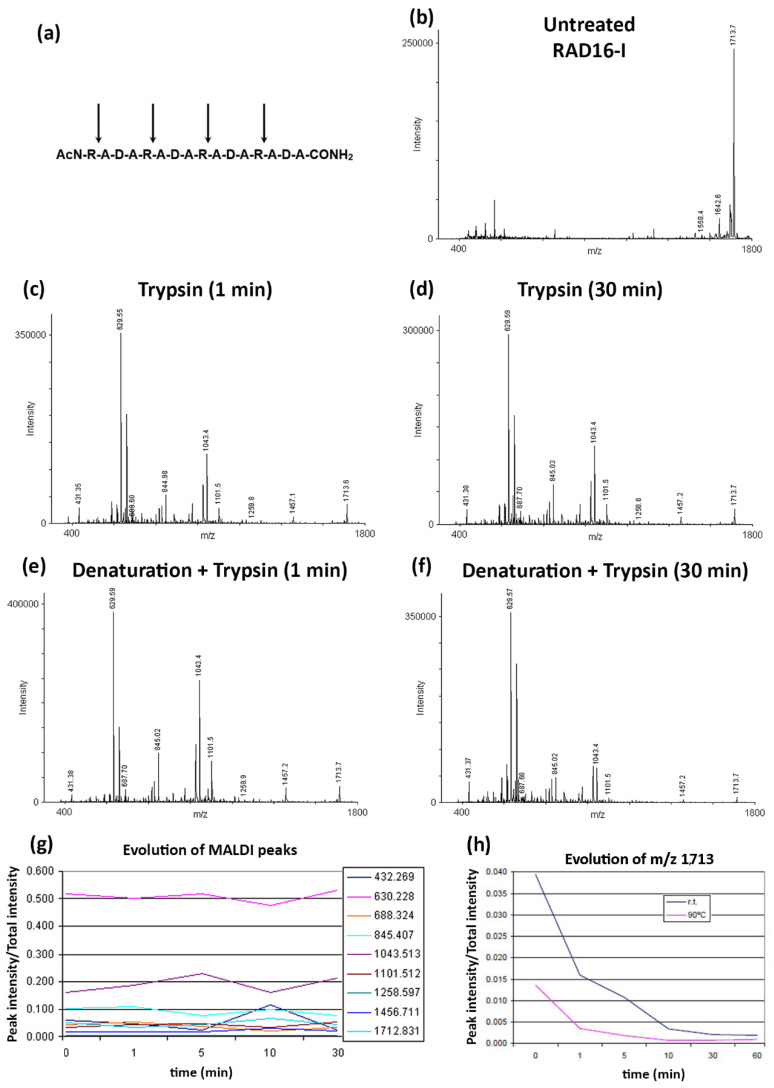
MALDI spectra of RAD16-I. (**a**) Potential cleavage sites of trypsin on RAD16-I; (**b**) MALDI spectrum of untreated RAD16-I; (**c**) MALDI spectrum of RAD16-I after 1 min trypsin incubation; (**d**) MALDI spectrum of RAD16-I after 30 min trypsin incubation; (**e**) MALDI spectrum of thermally denatured RAD16-I after 1 min trypsin incubation; (**f**) MALDI spectrum of thermally denatured RAD16-I after 30 min trypsin incubation; (**g**) Evolution of RAD16-I degradation peaks as a function of the incubation time with trypsin; (**h**) Evolution of RAD16-I peak (*m*/*z* 1713) in untreated and thermally treated trypsin digested samples inactivated by the MALDI-TOF sample preparation process.

**Figure 5 biomolecules-12-00411-f005:**
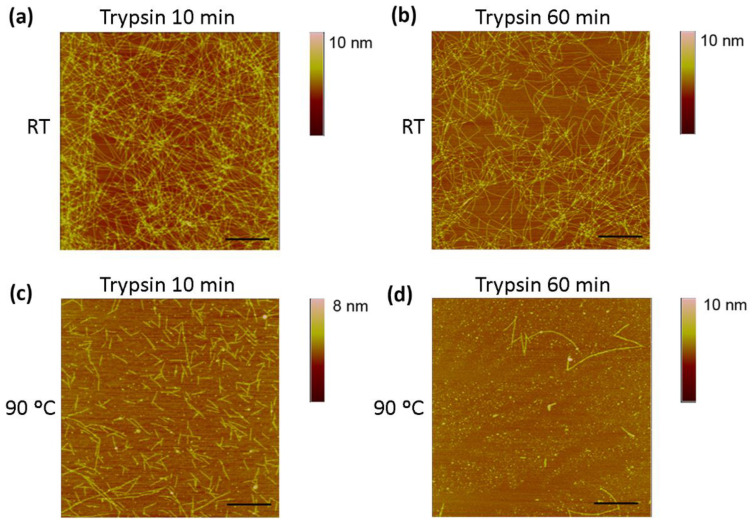
AFM images of untreated (RT) and thermally treated (90 °C) RAD16-I after trypsin incubation. RAD16-I incubated with trypsin for (**a**) 10 min and (**b**) 60 min. Thermally treated RAD16-I incubated with trypsin for (**c**) 10 min and (**d**) 60 min. Scale bars represent 500 nm.

**Figure 6 biomolecules-12-00411-f006:**
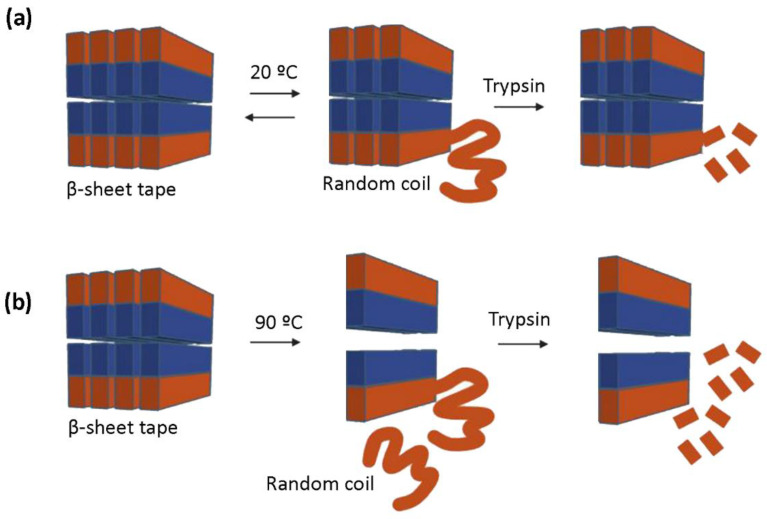
Schematic representation of the model for the tryptic degradation of untreated and thermally treated RAD16-I samples. (**a**) Terminally located peptides are in equilibrium between β-sheet and random coil structure; (**b**) Thermal treatment accelerates this structural transition, in which the random coil peptide molecules are dissociated from the fiber ending and, thus, becoming susceptible to trypsin cleavage. On the contrary, peptides located in the stable fiber structure are inaccessible to the enzyme.

**Table 1 biomolecules-12-00411-t001:** Peptide fragments that could theoretically be obtained after digestion of the oligopeptide RAD16-I with trypsin and its peak presence (+) or absence (−) in MALDI spectra.

Peptide Fragment	Molecular Weight	Peak Presence in MALDI Spectra
AcN-RADARADARADARADA-CONH_2_	1712	+
AcN-R	216	−
AcN-RADAR	629	+
AcN-RADARADAR	1043	+
AcN-RADARADARADAR	1456	+
ADA-CONH_2_	274	−
ADAR	431.4	+
ADARADARADARADA-CONH_2_	1514	−
ADARADARADA-CONH_2_	1102	+
ADARADA-CONH_2_	688	+
ADARADARADAR	1258	+
ADARADAR	844	+

## Data Availability

Data is contained within the article.

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
