# Peer review of "β-Sheet to Random Coil Transition in Self-Assembling Peptide Scaffolds Promotes Proteolytic Degradation"

_biomolecules, 2022, doi:10.3390/biom12030411_

Round 1

Reviewer 1 Report

Genové et al. reported the observation of the secondary structure transition in self-assembling RAD16-I peptide, which promotes proteolytic degradation. The story that the author presented is complete, and the English writing is professional and clear. However, I would recommend the authors to make necessary revisions, provide clarifications of the data, and further state the significance of the project:

(1) Since the trypsin digestion was very fast (in 30 min) for both thermally treated and untreated RAD16-I samples (Figure 4h), what is the significance of the observation that the 90 °C-treated peptides underwent faster trypsinization?

(2) Quantitative analysis based on MALDI signal intensity can be challenging and tricky, because the signal intensity may vary with sample composition, charge, morphology, laser exposure time, and laser conditions. If this is true, the analysis presented in Figure 4g and Figure 4h may not be reliable. That may also explain why some predicted peptide fragments listed in Table 1 were not detected on MALDI. Techniques that are performed in milder conditions and involve LC, such as LC-MS and LC-MS/MS, might be better ways to do the peptide mapping and confirms the efficiency of the trypsin digestion.

(3) Since the authors mentioned the potential of RAD16-I being used as biomedicine, how to make sure that there are trypsins available at the injury site or target organ? How to make sure the trypsin activity will not be affected by the cellular environment?

(4) Figure 3c: How to make sure the observation that shorter fibers were detected was due to the loss of self-assembling capacity of the thermally treated samples, not due to the high temperature degrading the fibril structures?

(5) For each AFM, MALDI, and CD experiment, please more clearly describe the timeline of the thermal treatment step and the trypsin digestion step. How long did each step take? In what order did the authors perform those steps?

Minor points:

(6) For the AFM part of the Method section, after rinsing twice with Milli-Q water, did you blot dry, blow dry, or air dry the water from the mica? Or did you perform the AFM with water on the mica? It would be nice to clarify this.

(7) It would be nice to include arrows/lines in Figure 3a and 3c, and connect them to Figure 3b and 3d, to indicate which regions were zoomed in. It also helps readers quickly understand that Figure 3b and 3d are zoomed-in views.

Reviewer 2 Report

The manuscript by Genove et al demonstrated the protease sensitivity of self-assembling amphiphilic peptides, in particular RAD16-I. It was shown that when RAD16-I is subjected to thermal denaturation, its secondary structure undergoes a transition from β-sheet to random coil.

There are many flaws in the paper which need to be addressed before publishing the manuscript.

The MALDI results for studying trypsin digestion are conducted very vaguely even the authors mentioning that at page 7 line 246. At page 7 line 251-253; it is unclear what author wants to describe.

The authors have observed the transition from beta-sheet to random coil at a temperature of 90oC. Upto 40oC they have observed no change in secondary structure. Did author study other temperatures between 40 to 90oC?

The secondary structure tradition is happening at a very high temperature; how this be useful considering practical application of RAD16-I as a biomaterial?

The study describes only one self-assembling peptide, it would be interesting to test this hypothesis on other self-assembling peptides as well for a comparison purpose. 

Reviewer 3 Report

Dear editor,

In this manuscript, combination of thermal degradation and proteolytic breakdown was used to study the RAD16-I peptide, which is known to self-assemble into fibril-like structures. It is shown that at peptide concentrations below ~1 mM, RAD16-I is unable to self-assemble into fibrils at high temperatures (i.e., 90o C) while it can do so at a concentration of 2.9 mM. Moreover, trypsin is shown to reduce significantly the amount of fibrils in the system for untreated and thermally treated samples in a time dependent manner. After 60 minutes of trypsin incubation, the amount of fibrils in the system is significantly less than after 1 minute trypsin incubation. An interesting schematic model is proposed to explain these observations.

This reviewer believes that results are convincing and could be of interest to the community studying self-assembly of peptides. However, a more extensive discussion of the literature related to thermal degradation of fibrils (which is critical to the current paper) is needed before he can recommend the manuscript. This reviewer also recommends including a discussion about the fibril-monomer equilibrium, which is central to the manuscript.

  1. In the manuscript, it is shown that for peptide concentrations below ~1 mM, RAD16-I does not self-assemble into fibril at high temperature. Several studies in the literature have been dedicated to understand the effect of temperature on fibril stability and the reviewer believes that they should be discussed. For example, the stability of fibrils was shown to be is strongly related to side chain interactions. For strongly hydrophobic peptides, fibrils become more stable with increasing temperature whereas for more polar sequence, the opposite trend is observed (see https://doi.org/10.3390/molecules24010202 and https://doi.org/10.1021/acs.jctc.9b00145 ). This was confirmed more recently by all-atom simulations wherein for amphipathic sequences where the non-polar residue is phenylalanine, leucine, or valine, fibrils become more stable with temperature. In contrast, for amphipathic sequences where the non-polar residue is alanine, fibrils become less stable with increasing temperature (see https://doi.org/10.1016/j.molliq.2021.118283 ). These studies support the finding in the manuscript for RAD16-I and they should be discussed.

2. The explanation of the experimental data is strongly related the fibril-monomer equilibrium in the system. The equilibrium has been studied experimentally for amyloid proteins (see    https://doi.org/10.1021/ar050069h ) and using theoretical modes of fibril formation (see https://doi.org/10.1016/j.bpc.2017.03.001 ). The insights that were gained from these studies are:

a) For a given concentration of fibrils in the system, the equilibrium concentration of RAD16-I monomers (C*) in the solution increases with temperature.

b) If cleavage of RAD16-I monomers by trypsin reduces the concentration of RAD16-I monomers in the solution below the equilibrium concentration (C*) then peptides at the extremity of the fibril will detach in an attempt to attain the equilibrium concentration-see above mentioned references. This explains why there is less fibrils in the system as the incubation time of trypsin increases. It also predicts that if the incubation time increases further (i.e., beyond 60 minutes), even less fibrils are expected to be in the system.

Round 2

Reviewer 1 Report

The authors have provided satisfactory explanations/answers to my comments and questions. As I suggested, they have addressed the significance of their study (line 85-96), a detailed timeline of their trypsin digestion (line 155-158), more details in the AFM studies (line 129, 133), and a more accurate figure caption (line 216-218).

My biggest concern was that MALDI is not a reliable way to do quantitative analysis. The authors have acknowledged that and claimed that MALDI analysis results should be (and have been) taken together with the AFM results. I accept that explanation and understand it is not practical and beyond the scope of this paper to perform quantitative LC/MS studies.

Overall, the revision is satisfactory, the manuscript is in good shape now, and I recommend the publication of this manuscript. 

Reviewer 2 Report

The author's have addressed majority of the concerns. I recommend publishing.